# Skin Inflammation and Testicular Function: Dermatitis Causes Male Infertility via Skin-Derived Cytokines

**DOI:** 10.3390/biomedicines8090293

**Published:** 2020-08-20

**Authors:** Ai Umaoka, Hiroki Takeuchi, Kento Mizutani, Naohiro Seo, Yoshiaki Matsushima, Koji Habe, Kohei Hagimori, Yukie Yamaguchi, Tomoaki Ikeda, Keiichi Yamanaka

**Affiliations:** 1Department of Dermatology, Mie University Graduate School of Medicine, 2-174 Edobashi, Tsu, Mie 514-8507, Japan; umaokaai@clin.medic.mie-u.ac.jp (A.U.); habe-k@clin.medic.mie-u.ac.jp (K.H.); 2Obstetrics and Gynecology, Mie University Graduate School of Medicine, 2-174 Edobashi, Tsu, Mie 514-8507, Japan; h-takeuchi@clin.medic.mie-u.ac.jp (H.T.); t-ikeda@clin.medic.mie-u.ac.jp (T.I.); 3Immuno-Gene Therapy, Mie University Graduate School of Medicine, 2-174 Edobashi, Tsu, Mie 514-8507, Japan; seo-naohiro@clin.medic.mie-u.ac.jp; 4Medicines Development Unit Japan, Eli Lilly Japan K.K., 5-1-28 Isogamidori, Chuo-ku, Kobe, Hyogo 651-0086, Japan; hagimori_kohei@lilly.com; 5Department of Environmental Immuno-Dermatology, Yokohama City University Graduate School of Medicine, Yokohama 236-0027, Japan; yui1783@yokohama-cu.ac.jp

**Keywords:** male infertility, dermatitis, inflammatory cytokine, sperm

## Abstract

The medical comorbidities including skin diseases are associated with male infertility. The most common cause of male infertility is the inability of testes to produce sperm; however, the influence of persistent dermatitis on testicular function has not been elucidated so far. We investigated the relationship between skin inflammation and impaired sperm production using a spontaneous dermatitis mouse model. We examined the breeding records of dermatitis mice and their wild-type littermates. Sperm count, motility, and viability were analyzed by direct microscopic observation and flow cytometry. In addition, testis and epididymis were histologically examined. Finally, sperm viability was evaluated in another dermatitis mouse model and in wild-type mice in which inflammatory cytokines were intraperitoneally administered. Compared to wild-type littermate mice, the number of children born was lower in mice with dermatitis. The body weight and testis size were decreased age-dependently. In the skin disease group, the sperm count and movement ratio were clearly decreased, and reduced sperm viability was observed. Histological examination revealed the detachment of Sertoli cells and reduced spermatogenesis. The fibrosis of epididymal stroma was severe, and it might affect defective sperm maturation in the epididymis. In addition, this phenomena was reproduced by a hapten applied dermatitis mouse model and the intraperitoneal administration of inflammatory cytokines. Once the skin is inflamed, inflammatory cytokines are produced and released, which may affect testicular and sperm function. Additional studies are needed to determine the relationship between male infertility and severe dermatitis in human.

## 1. Introduction

Infertility is a disease of the reproductive system defined by the failure to achieve a clinical pregnancy for 12 months or more with regular unprotected sexual intercourse [1], and it is becoming a widespread concern. Approximately half of infertility cases are due to male-related factors [2,3]. Increasing evidence has shown that male infertility is associated with an increased risk of common and incident tumors and cardiovascular, metabolic, and autoimmune diseases [4].

These processes may include genetic, developmental, or lifestyle-based factors; however, the exact nature of these associations remains unknown. The most common cause of male infertility is the inability to produce sperm in the testes, which occurs in approximately two-thirds of infertile men [4,5,6], and various studies have been conducted for sperm dysfunction [7,8], including the malformation of sperm heads [9,10], ciliopathy [8], and thyroid dysfunction [11], but the details of sperm dysfunction have not been determined. Focusing on skin diseases can be divided into congenital and acquired. Several genetic skin disorders are associated with hypogonadism as a prominent and consistent feature, such as De Sanctis-Cacchione syndrome, poikiloderma congenital, LEOPARD syndrome (progressive cardiomyopathic lentiginosis), and also ichthyosis including trichothiodystrophy and Werner syndrome. Acquired skin diseases include infection (leprosy and HIV), psoriasis, systemic lupus erythematosus, and zinc deficiency [12]. However, the causal relationship between severe dermatitis and male infertility has not been examined so far.

Therefore, we evaluated the influence of skin inflammation on sperm dysfunction in an acute spontaneous dermatitis mice model, keratin 14-specific human caspase-1-overexpressing transgenic (KCASP1Tg) mice [13]. We previously reported that inflamed skin produces and releases large amounts of pro-inflammatory cytokines including interleukin (IL)-1α/β compared to normal skin, which inflow into the systemic circulation, affecting distant organs. Systemic sclerotic changes of the large arteries represent aberrant remodeling of the vascular walls with aortic stenosis and the deterioration of peripheral circulation mimics human arteriosclerosis obliterans, which were partially ameliorated by the simultaneous treatment with neutralizing antibodies against IL-1 α⁄β [14], leading to the new concept of “inflammatory skin march” for psoriasis and atopic dermatitis [15]. Systemic amyloidosis with functional decline has also been observed in chronic inflammatory skin conditions. Histological examinations showed a dense amyloid and immunoglobulin G (IgG) deposition and loss of normal architecture in spleen, kidney, and liver [16]. In addition, KCASP1Tg mice gradually became emaciated as the dermatitis spreads. The supernatant from the homogenates of erupted skin lesions led to a severe reduction in produced lipid particles in an adipocyte culture system [14]. The supplementation of skin homogenate supernatant suppresses the production of lipid from adipocyte due to the overproduced inflammatory cytokines. In fact, computed tomography imaging of KCASP1Tg mice at 6 months of age revealed a dramatic decrease in visceral and subcutaneous fat compared to those in normal controls [14]. There had been only a few studies evaluating the relation between adipose tissue pathology and function in inflammatory skin disease. The skin-derived inflammatory cytokines led to the secretion of pro-inflammatory proteins from adipocytes and activated monocytes and lymphocytes infiltrated into the adipose tissue, which may lead to adipose tissue atrophy [17]. Thus, skin disease accelerates systemic inflammation. Then, we predicted that persistent dermatitis might deteriorate reproductive function, particularly testicular function. In the present study, we examined the breeding records, sperm count, sperm motility, sperm viability, and histopathology of testicular tissue in skin inflammation model mice and normal littermates. Finally, we measured sperm viability in another dermatitis model and wild-type mice in which inflammatory cytokines were intraperitoneally administered.

## 2. Materials and Methods

### 2.1. Mouse Study

We used spontaneous dermatitis model (KCASP1Tg) mice, which show acute erosive dermatitis starting at 8 weeks old. Male KCASP1Tg and littermate C57BL/6 control mice at ages of 8–32 weeks old were used. The experimental protocol was approved by the Mie University Board Committee for Animal Care and Use (#22-39-4, approval date was 03 December 2018). The difference in the number of children born from male KCASP1Tg and wild-type littermate male mice was evaluated as follows. The breeding method was a Harlem system of male KCASP1Tg or wild-type mouse: wild-type female mouse = 1:2. Mating was started at the age of 8 weeks, and 3 gauges were prepared for each group. The total number of pups born in 6 months was counted.

Oxazolone (OX, Sigma, St. Louis, MO, USA) was dissolved in acetone/olive oil (1:1). Seven-week-old male wild-type mice were initially sensitized by pasting 20 μL of 0.5% OX solution to their right ear 7 days prior to the first challenge (day –7) and then 200 μL of 0.5% OX solution was repeatedly applied on the back skin 3 times per week from day 0 until day 14. All mice were sacrificed on day 15. Six mice were used in this experiment.

Eight-week-old male wild-type mice were also treated by intraperitoneal injection of recombinant tumor necrosis factor-α (TNF-α), recombinant IL-1β, recombinant interferon-γ (IFN-γ), or phosphate-buffered saline (PBS, Nacalai tesque, Kyoto, Japan). The cytokines were diluted in PBS and administered at 250 µg/kg body weight. Injection was performed three times per week for two weeks. All recombinant proteins were purchased from Biolegend (San Diego, CA, USA). All mice were sacrificed on the day after the 6th injection. Ten mice were used in each group. In the IFN-γ group, 4 out of 10 died prematurely.

### 2.2. Tissue Sampling

The testis and epididymis from KCASP1Tg and wild-type mice were sampled when the mice reached ages of 8, 16, and 32 weeks. The sperm was removed from the epididymis and made into a suspension with 100 μL of warm modified human tubal fluid (HTF) Medium (FUJIFILM Irvine Scientific, Santa Ana, CA, USA). Ten μL of the suspension were used to determine the sperm count and motility rate using a Makler sperm count chamber (Alpharetta, GA, USA) under direct microscopic observation. Sperm movement was categorized based on the World Health Organization (WHO) laboratory manual on the Examination and processing of human semen 5th edition; motility was graded by distinguishing spermatozoa with progressive or non-progressive motility from those that were immotile. Progressive motility was considered as spermatozoa moving actively, either linearly or in a large circle, regardless of the speed. The sperm motility ratio was determined as the percentage of sperm with progressive motility among the whole sperm measured. One epididymis was measured per one mouse, and 10 mice were used in each group.

### 2.3. Flow Cytometry Analysis

The sperm count and viability were measured by flow cytometry (FCM). We sampled one epididymis per mouse and analyzed five mice in each group. The epididymal suspension was diluted by 20-fold with modified HTF medium containing 6% bovine serum albumin (Sigma-Aldrich, St Louis, MO, USA). The operations were carried out at 37 degrees centigrade, and we measured the whole samples. To evaluate the relationship between sperm viability and motility, the epididymal suspension was stained with thiazole orange (TO) solution (BD Biosciences, San Jose, CA, USA) to detect all cells and propidium iodide (PI, BD Biosciences, San Jose, CA, USA) staining solution to distinguish viable from non-viable sperm [18,19]. We added 2.0 μL of each dye solution into 2 mL of cell suspension. After vortex and incubation for 5 minutes at room temperature, the cell suspension was analyzed with an Accuri C6 (BD Biosciences, Franklin Lakes, NJ, USA). TO was 514/533 nm of excitation/emission maxima. TO was excited using a 488 nm laser paired with a 530/30 nm bandpass filter. On the other hand, PI was 538/617 nm of excitation/emission maxima. PI was excited using a 488 nm laser paired with a 585/40 nm bandpass filter. Five thousand events per replica were measured (*n* = 5, per each group).

### 2.4. Histological Analysis

The isolated testis and epididymis tissue were fixed in 10% formalin neutral buffer solution (Wako, Osaka, Japan) for 12 h, dehydrated in an ethanol series (Wako), embedded in paraffin, cut into 6-µm sections, and stained with hematoxylin and eosin (HE), Masson trichrome (Muto Pure Chemicals, Tokyo, Japan), anti-hypoxia inducible factor-1α (HIF1α, Sigma-Aldrich, St. Louis, MO, USA) and anti-mouse IgG specific monoclonal antibody (Abcam, Eugene, OR, USA).

### 2.5. The Cytokine Concentration in the Testis

The cytokine concentration was analyzed in the epididymal suspension at the age of 16 weeks of KCASP1Tg and wild-type mice. The concentrations for TNF-α, IL-1β, IL-12p70, MCP-1, IFN-γ, IL-6, KC, IL-10, IL-1α, IP-10, and IL-23 were measured using an AimPlex™ multiplex assay: mouse inflammation 11-Plex kit (Aimplex Bipscinece, St. Louis, MO, USA), and analyzed by FCM. This technology utilizes multiple bead populations differentiated by size and different levels of fluorescence intensity. With two sizes of beads and 12 levels of fluorescence intensity in each bead size, the bead populations in the reaction are determined using a standard FCM. Concentrations of the protein of interest in the samples can be obtained by comparing the fluorescent signals to those of a standard curve generated from a serial dilution of a known concentration of the protein analyzed.

### 2.6. Statistical Analysis

Statistical analysis was performed by using GraphPad Prism software version 8 (GraphPad, Inc. San Diego, CA, USA). Two group comparison was analyzed by Mann–Whitney test and three or more group comparisons were conducted by the Kruskal–Wallis test. For the analysis of the number of pups born, we used a chi-square test. Differences were considered significant when was *p* < 0.05.

## 3. Results

### 3.1. Number of Pups Born

We employed spontaneous dermatitis model (KCASP1Tg) mice, which showed acute erosive dermatitis starting at 8 weeks old. The male KCASP1Tg or wild-type mice were bred with a Harlem system as described in the material and method section, and mating was started at the age of 8 weeks. In 6 months, 81 pups were born from KCASP1Tg mice (male and female = 37:44) and 113 were born from wild-type mice (male and female = 62:51). The relative frequency of the male–female ratio was 0.46:0.54 for KCASP1Tg mice and 0.55:0.45 for wild-type mice. The statistic was not significant with the chi-square test (*p* = 0.26).

### 3.2. Characteristics of Phenotype and Testis

From 8 weeks of age, KCASP1Tg mice began to scratch their bodies, and erosive rashes appeared from the face, covering approximately 20% of the body surface when the mouse was 16 weeks old. Along with the expansion of eruption, the body weight decreased compared to their wild-type littermates. The mean ± SD of body weight (g) was at 8 weeks of age (19.9 ± 0.99, 20.4 ± 1.07), 16 weeks of age (21.8 ± 1.69, 27.7 ± 1.57), and 32 weeks of age (21.1 ± 2.38, 29.8 ± 1.55) for KCASP1Tg mice and wild-type littermate mice, respectively. A significant difference was observed at 16 and 32 weeks (*p* < 0.0001 for both groups, Figure 1A,B). In contrast, the testis size and weight were significantly different between KCASP1Tg and wild-type littermate mice at the age of 32 weeks (*p* = 0.0009). The mean ± SD of the weight of testis (g) was at 8 weeks of age (0.0883 ± 0.00800, 0.0918 ± 0.01117), 16 weeks of age (0.0857 ± 0.01121, 0.0983 ± 0.01378), and 32 weeks of age (0.0604 ± 0.02126, 0.1026 ± 0.01367) for KCASP1Tg mice and wild-type littermate mice, respectively (Figure 2A,B).

### 3.3. Sperm Count and Sperm Motility

The sperm was purified from the epididymis of KCASP1Tg and wild-type mice at 8, 16, and 32 weeks. A video of sperm movement was recorded at 8, 16, and 32 weeks of wild-type (Figure 3A) and KCASP1Tg mice (Figure 3B). The number of sperm was low, and movement was slow in KCASP1Tg mice. The sperm count was measured with two different systems. Using a Makler sperm count chamber under direct microscopic observation, the sperm count was significantly decreased at 16 and 32 weeks in KCASP1Tg mice (*p* = 0.0232 and *p* < 0.0001, respectively). The mean ± SD of the sperm count (×10^4^) was at 8 weeks of age (53.7 ± 41.66, 78.3 ± 35.91), 16 weeks of age (34.2 ± 33.47, 82.4 ± 42.25), and 32 weeks of age (3.9 ± 5.10, 53.0 ± 50.34) in KCASP1Tg mice and wild-type littermate mice, respectively (Figure 4A). In contrast, the sperm count was decreased at 8 and 16 weeks in KCASP1Tg mice (*p* = 0.0079 and *p* = 0.0317, respectively) according to flow cytometry (FCM) analysis. The mean ± SD of the sperm count (×10^4^) by FCM analysis was as follows: 8 weeks of age (21.1 ± 8.80, 53.0 ± 4.05), 16 weeks of age (14.7 ± 12.94, 44.8 ± 16.17), and 32 weeks of age (12.4 ± 9.03, 19.1 ± 5.37) in KCASP1Tg mice and wild-type littermate mice, respectively (Figure 4B). Sperm movement was graded by distinguishing spermatozoa with progressive or non-progressive motility from those that were immotile. The sperm motility ratio was significantly decreased at 32 weeks in KCASP1Tg mice compared to their wild-type littermates (*p* = 0.0031). The mean ± SD of the sperm motility ratio was at 8 weeks of age (46.8 ± 19.01, 50.5 ± 8.07), 16 weeks (38.6 ± 19.96, 46.9 ± 15.04), and 32 weeks (7.1 ± 11.03, 34.7 ± 18.14) in KCASP1Tg mice and wild-type littermate mice, respectively (Figure 4C). These alterations may be indicative that the sperm concentration was low and/or alterations in spermatogenesis and/or sperm maturation in KCASP1Tg mice.

### 3.4. Sperm Viability

The sperm viability was measured by FCM to distinguish viable and non-viable sperm. The mean ± SD of the sperm viability (%) was at 8 weeks of age (30.8 ± 11.33, 70.0 ± 5.27), 16 weeks of age (27.9 ± 9.83, 45.0 ± 6.84), and 32 weeks of age (25.3 ± 17.58, 39.7 ± 5.19) in KCASP1Tg mice and wild-type littermate mice, respectively. The sperm viability was decreased at 8 and 32 weeks in KCASP1Tg mice (*p* = 0.0079 and *p* = 0.0482, respectively, Figure 5A,B). The current result suggested that the sperm of KCASP1Tg mice were more likely to die than those of wild-type mice.

### 3.5. Histological Analysis of Testis and Epididymis

Hematoxylin and eosin (HE) staining of the testis of 32-week-old KCASP1Tg mice showed the stagnation of spermatogenesis in the seminiferous tubules. On the other hand, no morphological change of the seminiferous epithelium was observed, and many spermatozoa were found in the lumen from 2 to 8 months in wild-type mice (Figure 6A, HE, ×200).

We also observed the vacuolation of Sertoli cells, syncytia formation, lack of spermatogenesis, and incorrectly rearranged seminiferous epithelium in 32-week-old KCASP1Tg mice compared to wild-type littermates. Leydig cells in KCASP1Tg mice were atrophied (Figure 6A, HE, ×400). Differences of fibrogenesis around seminiferous tubules and interstitial connective tissue were not clear in KCASP1Tg mice testis compared to wild type at any ages by Masson trichrome staining (Figure 6A). Hypoxia inducible factor-1 (HIF-1) staining showed no significant change in mice at all ages analyzed. IgG was intensely stained around the seminiferous tubules in the testis of KCASP1Tg mice at 32 weeks compared to younger mice.

In 8-week-old wild mice epididymis, spermatozoa were present throughout the caput, corpus, and cauda epididymis, and the maturation of spermatozoa in the epididymis was observed (HE, ×40, ×400). On the other hand, less sperm were observed throughout the caput, corpus, and cauda epididymis in 32-week-old KCASP1Tg mice compared to wild-type mice at 8, 16, and 32 weeks and KCASP1Tg mice at 8 and 16 weeks. The increased fibrogenesis of ductus epididymidis and connective tissues was detected by Masson trichrome staining in 32-week-old KCASP1Tg mice (Figure 6B).

### 3.6. Sperm Viability by Repeated Hapten Application and Intraperitoneal Injection of Recombinant Proteins

Oxazolone (Ox) is one of the haptens, and it is commonly used to induce allergic contact dermatitis. In the OX-applied dermatitis model, the viable sperm was significantly decreased compared to wild-type controls measured by FCM (*p* = 0.0002). The administration of inflammatory cytokines including recombinant interferon-γ (INF-γ), tumor necrosis factor-α (TNF-α), and IL-1β into wild-type mice revealed decreased sperm viability compared to that in phosphate-buffered saline (PBS)-injected mice. The mean ± SD of the viable cells (%) of wild-type mice, OX-applied mice, INF-γ-, TNF-α-, and IL-1β-treated mice were 57.0 ± 4.77, 30.0 ± 11.7, 45.3 ± 6.07, 42.8 ± 10.52, and 40.4 ± 8.17, respectively. The administration of inflammatory cytokines resulted in decreased sperm viability compared to in the PBS-injected mice (*p* = 0.0017, *p* = 0.0185, and *p* = 0.0007, respectively); however, the difference was not statistically significant among the different cytokines administered (Figure 7).

### 3.7. The Cytokine Concentration in the Testis

Finally, the cytokine concentration in the testis was measured at 16 weeks. The concentrations for TNF-α, IL-1β, IL-12p70, MCP-1, IFN-γ, IL-6, KC, IL-10, IL-1α, IP-10, and IL-23 were measured using FCM. Between KCASP1Tg and wild-type mice, INF-γ, KC (a mouse neutrophil chemoattractant protein), and IP-10 (CXCL10 chemokine, produced from monocyte, endothelial cell, and fibroblast when treated with IFN-γ) were significantly different (*p* = 0.008, *p* = 0.015, and *p* = 0.032, respectively). TNF-α was undetected in the current system. IL-1α and IL-1β in KCASP1Tg mice showed an increasing tendency compared to wild-type mice, but the difference did not reach significance (Figure 8). No significance was detected for other cytokines.

## 4. Discussion

The testis is an immunologically isolated organ of the body and is unlikely to cause inflammation [20]. One of the reasons is that Sertoli cells are believed to favor local immune tolerance against testicular autoantigens by segregating the autoantigens because of the blood–tubular barrier and by secreting immunosuppressive factors [21]. Our results showed that severe dermatitis leads to multiple alterations of spermatogenesis and sperm viability.

We characterized the role of skin-derived inflammatory cytokines in reproductive function, particularly testicular function. Medical comorbidities are associated with impaired sperm production [22], and skin disease is one of the suggested comorbidities. Our results showed the decreased sperm count and sperm motility ratio in dermatitis mice compared to wild type, suggesting the sperm does not function properly under inflammatory skin conditions, leading to the reduced number of births.

The itching is often accompanied in patients with atopic dermatitis (AD) and induces intensive scratching of the skin lesions. The eczematous inflammation releases a massive amount of pro-inflammatory cytokines stored in the epidermis into the systemic circulation targeting the distant organs. KCASP1Tg mice continually discharge IL-1α/β and other cytokines from the eczematous skin lesions, resulting in the non-aterotic aortic sclerosis, and this arteriosclerosis was perticially ameliorated by the combination treatment with anti-IL-1α/β neutralizing antibodies. In psoriasis, the keratinocyte-derived cytokines have been speculated as the causable factors for the ‘‘psoriatic march’’ causing myocardial infarction and cerebrovascular disorders. IL-1α/β are one of the major inflammatory cytokines from both AD and psoriasis skin lesions, leading to a new concept of ‘‘inflammatory skin march’’. From these reason, we proposed the connection between skin disorders and testicular toxicity.

The number of pups born in 6 months from KCASP1Tg mice was 70% of that from wild-type mice. KCASP1Tg mice begin scratching their bodies, and rashes appeared from the face and spread to the whole body. Along with the expansion of dermatitis, the body weight was decreased compared to wild-type littermates. These differences became apparent at 16 weeks and noticeable at 32 weeks (Figure 1A,B). This has been explained by adipose tissue atrophy in the body fat and subcutaneous fat due to increased levels of circulating inflammatory cytokines derived from inflamed skin [14] and also due to the interaction between adipocytes and infiltrating inflammatory cells into the adipose tissue [17]. In fact, the infiltrating monocytes and lymphocytes showed increased surface activation markers and adipocytes were small and irregularly shaped: the so-called burn-out phenotype [14,17]. On the other hand, the testis was small and hard, and the color was deep orange in KCASP1Tg mice (Figure 2A,B). We next analyzed sperm motility. The video record of sperm revealed that the sperm count was low and the sperm showed sluggish movement in KCASP1Tg mice compared to that in their wild-type littermates (Figure 3), which was evident at 16 weeks old and more prominent at 32 weeks old. The sperm count measurment using two different approaches was similar, and the sperm motility ratio was clearly decreased at 32 weeks in KCASP1Tg mice (Figure 4). Then, we investigated the sperm viability with sensitive staining using FCM. Live cells have intact membranes and are impermeable to dyes such as propidium iodide (PI), which only leaks into cells with compromised membranes. Thiazole orange (TO) is a permeant dye and enters into live and non-viable cells to varying degrees. Thus, a combination of these two dyes provides a rapid and reliable method for discriminating live and non-viable cells. Other candidates are SYBR green, which enters into living cells, and Hoechst 33,342 or 33,258, which stain both live and dead cells. Therefore, the SYBR/PI method and Hoechst/PI method are other choices. The TO/PI method is one of these options, and it is used in various animal species [18,19]. The sensitive staining revealed that the KCASP1Tg mice showed a significantly decreased sperm viability ratio compared to their wild-type littermates, and this may be critical for the fertilization (Figure 5). Sperm are very delicate and are directly affected by the surrounding environments. Therefore, the clinical effects of skin inflammation or condition on the fertility may depend on the age of diseases onset, duration, and the severity of the skin inflammation. It is necessary to take statistics by human analysis.

Sertoli cells were detached from the basement membrane and the vacuolation of Sertoli cells; syncytia formation were detected. The fewer growing sperms were observed in the seminiferous tubules at 32 weeks of age in KCASP1Tg mice (Figure 6A). As reported previously, high serum IgG and IgG deposition in various organs have been detected in KCASP1Tg mice [16]. IL-1α/β is an essential cytokine for B cell activation and the production of immunoglobulin from B cells [23,24]; therefore, increased IL-1α/β levels derived from inflamed skin may influence the B cell maturation, activation, and IgG production in the spleen. This intense IgG deposition finally leads to a loss of normal architecture in several organs. In the current study, IgG was already deposited at 8 weeks, and it strongly stained around the seminiferous tubules at 32 weeks of KCASP1Tg mice, indicating that spermatogenesis may have been affected (Figure 6A). The nutritional supply or circulation itself may be hampered by the dense IgG deposition. In autoinflammatory disorders and autoimmune diseases, plasma IgG levels are often increased clinically; therefore, it may be important to analyze the sperm quality in patients with comorbidities. Another reason may be the direct influence of a high concentration of plasma cytokines to Sertoli cells and Leydig cells. Leydig cells release several classes of hormones, including testosterone and androstenedione, and they were atrophied in KCASP1Tg mice. The size of the testis was decreased in an age-dependent manner in KCASP1Tg mice, which likely occurred because of fibrogenesis of the epididymis tube matrix, as shown by Masson trichrome staining. A high magnification of HE staining of the lumen of the seminiferous tubule showed the enlarged denatured sperm at 32 weeks of KCASP1Tg mice. The sperm maturation seemed to be hampered in KCASP1Tg at 32 weeks of age through caput, corpus, and caudal epididymal regions. Furthermore, there seemed to be less stored sperm in the epididymal caudal (Figure 6B). The strong fibrogenesis was observed in the ductus epididymidis and connective tissues, and environments are not preferable for the stored sperm, but the main reason for less sperm in the epididymis seems to be lowered sperm production in the testis.

The skin is one of the most critical immune organs, featuring skin-associated lymphoid tissues [16]; once irritated or activated, the skin produces and releases large amounts of inflammatory cytokines, which affect internal organs. We tested the sperm viability in two additional protocols. In the OX application in another dermatitis model, reduced sperm viability was also observed (Figure 7). IFN-γ injection into mouse testis induced impaired spermatogenesis, demonstrating that IFN-γ induced testicular toxicity [25,26]. A more recent study showed that the Fas system in Sertoli cells has been implicated in the maintenance of the immune privilege in the testis as well as in the regulation of germ cell apoptosis. Sertoli cells treated with TNF-α and/or IFN-γ increased their expression of Fas Ag both in membrane-bound and soluble form. Moreover, following cytokine stimulation, Sertoli cells become susceptible to Fas-mediated cytotoxicity, which suggests the role of inflammatory cytokines on Fas system regulation in the seminiferous epithelium [27,28]. The direct major cytokine administration into the abdomen resulted in the decreased sperm viability (Figure 7), which may be due to Sertoli cells apoptosis via the Fas-mediated way. Unexpectedly, the significance was undetected for sperm viability between the administrated cytokines, suggesting that all cytokines increased the risk of sperm abnormalities and may have additive effects.

KC and IP-10 are produced from the epithelial cells and fibroblast, and the production is enhanced by IFN-γ stimulation [29]. In our study, the cytokine concentration in the testis at the age of 16 weeks was significantly different between KCASP1Tg and wild-type mice for INF-γ, KC, and IP-10. KC and IP-10 are inflammatory chemokines and chemoattractant proteins, and they may have a direct effect on the spermatogenesis. TNF-α was undetected, and although the difference did not reach significance, there was an increasing tendency for IL-1α and β in KCASP1Tg mice compared to wild-type mice (Figure 8). The exact cause of the increased cytokine concentration in the testis was unclear, but circulating blood cytokine levels might influence it or the result in the fibrogenesis and degeneration of stroma. The testis in the inflammatory skin conditions is not a favorable and healthy environment, and the elevated cytokine levels may affect the sperm dysfunction.

Male infertility may be associated with unhealthy conditions [4]. Skin diseases may affect male fertility at various levels: pretesticular (hypothalamic/pituitary), testicular, and post-testicular (genitourinary ducts/accessory sex organs). Furthermore, general and systemic diseases may affect the alterations of semen quality that include reduced volume, sperm concentration, sperm motility, and sperm morphology. Clinical management of the skin diseases may be considered not only for the skin involvement but also for the reproductive system [12]. In the current study, we examined the relationship between infertility and skin inflammation; however, additional studies are needed to determine the potential mechanisms and further clarify the relationship between male infertility and severe dermatitis in human.

## Figures and Tables

**Figure 1 biomedicines-08-00293-f001:**
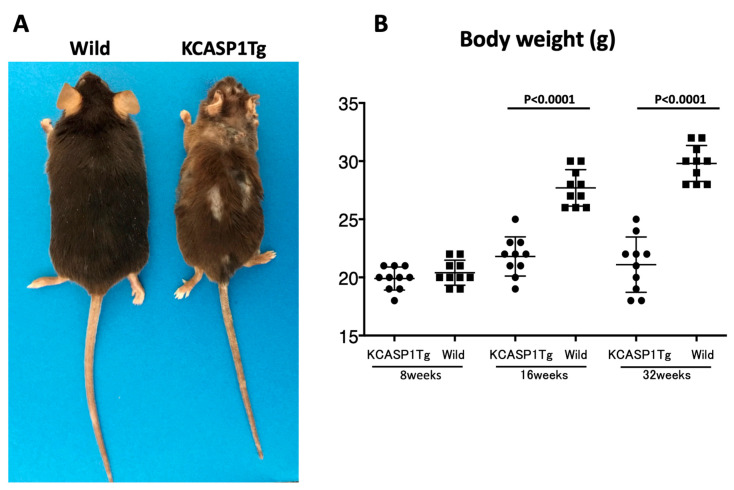
Phenotype and body weight. KCASP1Tg (keratin 14-specific human caspase-1-overexpressing transgenic) mice showed the scratching behavior and erosive dermatitis spread from the face to back skin. The clinical phenotype is similar to that of acute phase atopic dermatitis. The body of KCASP1Tg mice was smaller than that of wild-type littermate mouse at 16 weeks of age (**A**). We compared body weight between KCASP1Tg mice and wild-type littermates. The difference in body weight between the two groups was significant at 16 and 32 weeks of age (*p* < 0.0001 for both groups, **B**).

**Figure 2 biomedicines-08-00293-f002:**
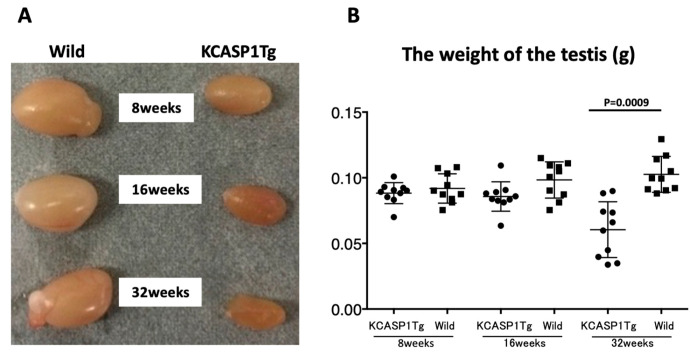
Clinical picture and weight of the testis. Testis size was smaller in KCASP1Tg mice than in wild-type mice and was dependent on the age. The color was dark orange for 16 and 32 weeks age of KCASP1Tg mice (**A**). Testis weight was compared between KCASP1Tg mice and their littermate controls, which was lower in KCASP1Tg mice than in wild-type mice with significance at the age of 32 weeks (*p* = 0.0009, **B**).

**Figure 3 biomedicines-08-00293-f003:**
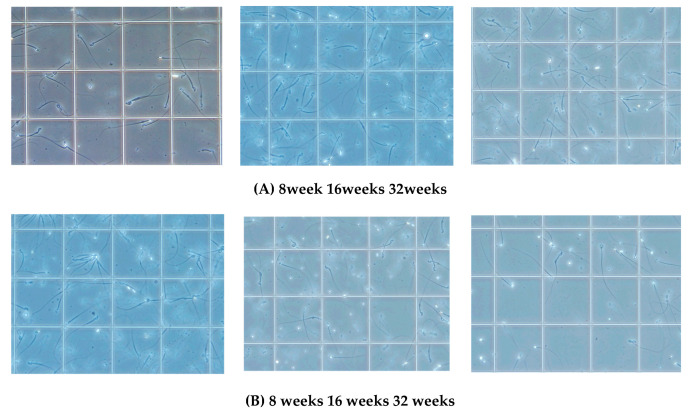
Video of sperm movement (×200). Video records of sperm movement at 8, 16, and 32 weeks of age of wild-type mouse (**A**, upper movie, double click to start) and KCASP1Tg mouse (**B**, lower movie) were shown. The number of sperm was low, and movement was slow in KCASP1Tg mouse.

**Figure 4 biomedicines-08-00293-f004:**
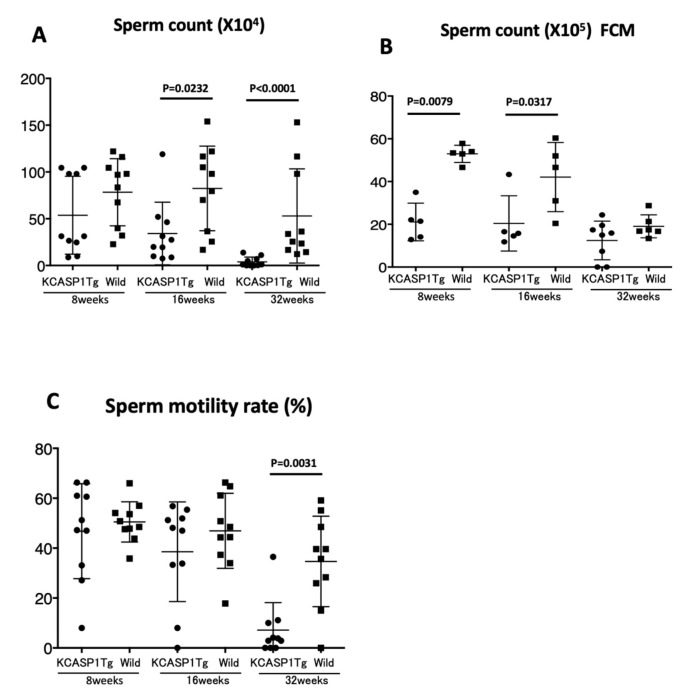
Sperm count and sperm motility. Sperm count was measured with two different methods. By using a Makler sperm count chamber under the direct microscopic observation, a significant decreased sperm count was detected at 16 and 32 weeks of KCASP1Tg mice (*p* = 0.0232 and *p* < 0.0001, respectively, **A**). Flow cytometry (FCM) analysis also revealed a decreased sperm count at 8 and 16 weeks of KCASP1Tg mice (*p* = 0.0079 and *p* = 0.0317, respectively, **B**). The sperm motility ratio was significantly decreased at 32 weeks of KCASP1Tg mice compared to wild-type controls (*p* = 0.0031, **C**).

**Figure 5 biomedicines-08-00293-f005:**
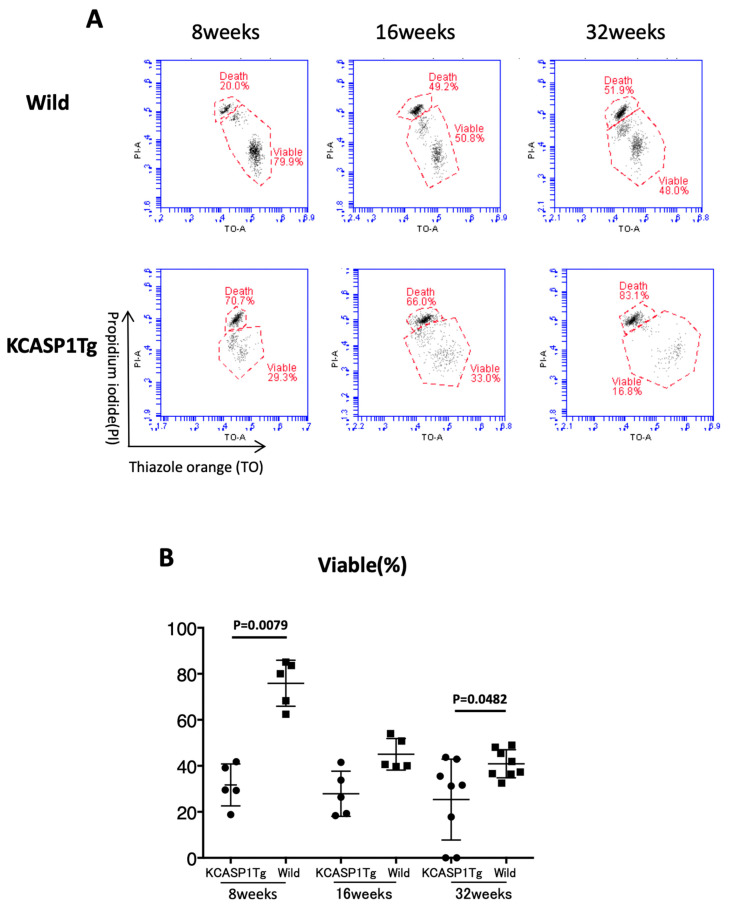
Sperm viability. The representative FCM data for the percentages of viable sperm cells at 8, 16, and 32 weeks of age in KCASP1Tg mice and wild-type mice are shown. Wild-type mice at 8 and 32 weeks of age showed a significantly larger number of viable sperm than KCASP1Tg mice (*p* = 0.0079 and *p* = 0.0482, respectively, **A**, **B**).

**Figure 6 biomedicines-08-00293-f006:**
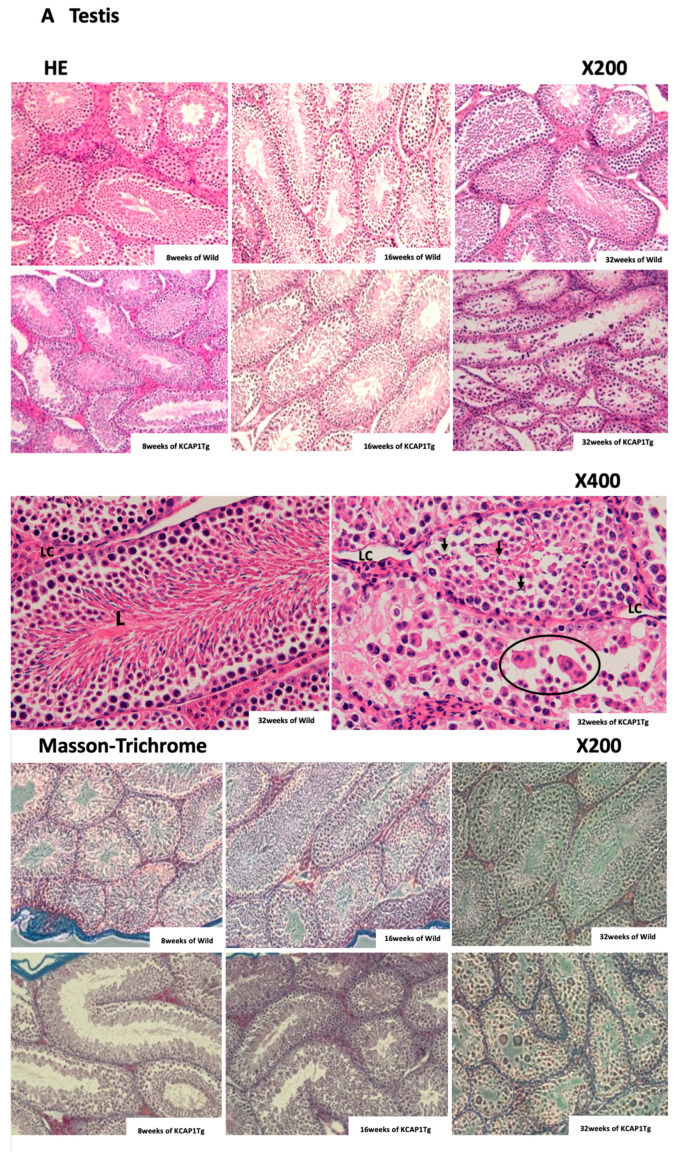
Histological findings for 8, 16, and 32 weeks of age of testis and epididymis in KCASP1Tg and wild-type mice. Histological changes of testis (**A**) and epididymis (**B**) in KCASP1Tg and wild-type mice at 8, 16, and 32 weeks of age were shown (×200, ×400). Comparison of the seminiferous tubules cross-sections, KCASP1Tg mice at 32 weeks old showed reduced sperm content in the lumen (L) of seminiferous tubules (×200). Large round cells with abnormal mitosis (circled) and sperm head (arrow) were seen in the lumen in KCASP1Tg mice, but not in the wild type. Leydig cells (LC) in KCASP1Tg mice were atrophied compared to wild type (×400). The vacuolation of Sertoli cells, syncytia formation, and incorrectly rearranged seminiferous epitheliums were observed in KCASP1Tg mice compared to wild-type littermates of normal tubule cross-sections (×400, **A**). Difference in the fibrogenesis of seminiferous tubules and interstitial connective tissue was not clear for both groups by Masson trichrome staining of the testis. Hypoxia inducible factor-1 (HIF-1) staining showed no remarkable difference at all ages between the two groups. Immunoglobulin G (IgG) deposition was increased in the stroma surrounding the seminiferous tubules of KCASP1Tg mice, even in mice at 8 weeks of age, but especially at 32 weeks old. Compared to the wild-type mice at 8 weeks of age, the sperm seems not to be mature in KCASP1Tg at 32 weeks of age through caput, corpus, and caudal epididymal regions (×40). Fibrogenesis was observed intensely in the ductus epididymidis and connective tissues by Masson trichrome staining (×200). S means sperm (**B**).

**Figure 7 biomedicines-08-00293-f007:**
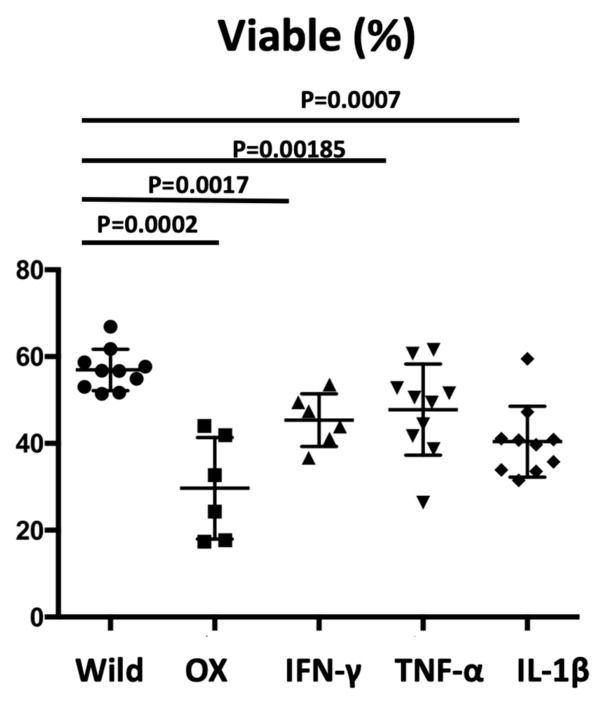
Sperm viability by repeated hapten application and intraperitoneal injection of recombinant proteins. In the oxazolone applied dermatitis model, the sperm viability was significantly decreased (*p* = 0.0002). On the other hand, intraperitoneal administration of inflammatory cytokines, recombinant interferon-γ (INF-γ), tumor necrosis factor-α (TNF-α), and interleukin (IL)-1β resulted in the decreased sperm viability compared to phosphate-buffered saline (PBS) injection (*p* = 0.0017, *p* = 0.0185, and *p* = 0.0007, respectively). However, no significant difference between each cytokine administered was detected.

**Figure 8 biomedicines-08-00293-f008:**
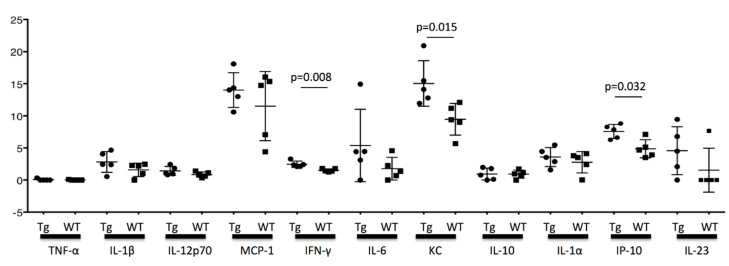
Cytokine concentration in the testis. Cytokine concentration in the testis was measured using FCM at the age of 16 weeks. The significant difference between KCASP1Tg and wild-type mice was detected for INF-γ, KC (a mouse neutrophil chemoattractant protein), and IP-10 (CXCL10 chemokine, produced from monocyte, endothelial cell, and fibroblast when treated with IFN-γ) (*p* = 0.008, *p* = 0.015, and *p* = 0.032, respectively). TNF-α was undetected. No significance was detected for other cytokines.

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
