# Peer review of "Skin Inflammation and Testicular Function: Dermatitis Causes Male Infertility via Skin-Derived Cytokines"

_biomedicines, 2020, doi:10.3390/biomedicines8090293_

Round 1
Reviewer 1 Report
General comments
Authors have corrected the manuscript according to most of the previous comments. Nevertheless, the quality of all the sections of the manuscript is still low. Therefore, authors must carefully revise the Results section in order to improve its readability; they also must use the past form in the whole section. The Discussion section still contains several paragraphs repeating the results obtained and not providing any clue about their significance. Finally, I would like to note that the authors did not revise and correct the Abstract section in accordance with the modifications performed in the text, and it contains serious mistakes.
Specific comments
Abstract
- Line 23. Please, also indicate that the sperm viability was also measured
- Line 29. Please correct “an increased number of dead sperms” by “reduced sperm viability”
- Line 30. Sperm maturation does not take place in the seminiferous tubules but in the epididymis. Authors must correct this sentence by using “disrupted spermatogenesis” instead of “deterioration of sperm maturation in the seminiferous tubules”.
- Line 31. “Fibrosis of the stroma”. Which stroma did the authors refer? Testicular stroma, epididymal stroma or both? Please, specify.
- Line 31. “destructed spermatogenesis was found in the epididymis”. As indicated before, the spermatogenesis takes place in the seminiferous tubules, whereas the sperm maturation occurs in the epididymis. So, authors must correct “denaturated spermatogenesis” by “defective sperm maturation”. By the way, how did the authors analyse the epididymal sperm maturation? The procedure used is not described in the Material and Methods section. Therefore, authors must be careful with the terms used.
- Lines 32-33. This sentence is unclear, considering that in fact the authors are confuse about where the spermatogenesis and sperm maturation occur.
- Lines 36-37. Authors must revise this conclusion because it does not seem obvious from their results.
Introduction
- Lines 50-52. Which the message? How is this related with the manuscript? Please, revise this sentence in order to provide a clear message according with the topic of the manuscript.
- Lines 79. Please, correct “testicular toxicity” by “testicular function”.
- Line 80. The authors also analysed the sperm viability. So, it must be added in this sentence.
Results
- As indicated in the general comments, this section must be carefully revised in order to improve its readability; authors must also use the past form in the whole section.
- Lines 153-155. The content of these lines does not belong to the Results section, and they must be eliminated. Please, not that the term “dead sperm” is not correct and authors must refer as “non-viable sperm” when describing the procedure to analyse the sperm viability. Authors should revise the whole document in order to unify the terminology.
- Line 162. “Sperm viability rate”. Did the authors measure the sperm viability rate? If so, specify how it was calculated. Otherwise, please just refer to “Sperm viability”.
- Lines 164-165. It is not necessary to indicate in the Figure legend the statistical test used. So, the content of this line must be removed.
- Line 172. “no clear change”. What do the authors refer? Please, provide a comprehensive description of the results.
- Lines 180-181. “normal maturation process”. How was it evaluated? Please, give some details about the sperm maturation process in wild type mice.
- Line 181-182. What do the authors refer when saying that “the sperm stored in the epididymis at 32-week-old 181 KCASP1Tg mice was few”. Did they perform any comparisons with wild type males?
- Line 191. Please, eliminate “The” from the beginning of the sentence.
- Line 192. Please, correct “Histopathological” by “Histological”. Please, note that the term histopathological is incorrect when referring to wild type samples.
- Lines 193-200. The information given in these lines is the same as in the results, so it must be improved. Please, note this suggestion was already given in the previous revision.
- Lines 202 and 205. Please, correct “live cell” by “viable sperm”. Authors must uniform the criteria in the whole document.
- Lines 216, 217 and 219. Please, correct “Sperm viability ratio” by “Sperm viability”, since the authors did not calculate this ratio.
- Line 221. Please, eliminate the last sentence of the figure legend.
- Lines 224-234. Please, revise the message of these lines in order to provide a clear description of the results obtained. Moreover, avoid starting sentence with “There was”
- Line 236. Please, eliminate “The” from the title of the figure.
- Lines 241. Please, eliminate the last sentence from the Figure legend.
Discussion
- Some paragraphs of the Discussion are a mere repetition of the Results section; a deep analysis of the significance of testis and epididymis pathologies, as well as of the sperm quality alterations is still lacking.
- Lines 247-248. Authors must revise the content of these two lines since they did not analyse the semen quality, but the spermatogenesis and sperm maturation. Moreover, the “sperm density” is not a sperm parameter.
- Line 276. “Sperm function”. The authors did not analyse the sperm function in their study. I suppose they refer to “sperm motility”. If so, please use appropriate terminology.
- Again, “sperm density” is not a sperm parameter. Do the authors refer to sperm concentration? Please, use appropriate terminology.
Material and Methods
- Subsection 4.3. Flow cytometry analysis. Authors must indicate: 1) which type of filters did they use to detect the fluorescence emitted by each fluorochrome, 2) how many replicas per sample did they analyse, 3) how many events were analysed by replica.
- Subsection 4.3. Flow cytometry analysis. Please, describe the protocol used to analyse the sperm viability.
- Line 382. Correct “dead sperm” by “non-viable sperm”.
Author Response
General comments
Authors have corrected the manuscript according to most of the previous comments. Nevertheless, the quality of all the sections of the manuscript is still low. Therefore, authors must carefully revise the Results section in order to improve its readability; they also must use the past form in the whole section. The Discussion section still contains several paragraphs repeating the results obtained and not providing any clue about their significance. Finally, I would like to note that the authors did not revise and correct the Abstract section in accordance with the modifications performed in the text, and it contains serious mistakes.
Response: We appreciated your comments. We have revised the abstract and text as you pointed.
Specific comments
Abstract
- Line 23. Please, also indicate that the sperm viability was also measured
Response: Thank you very much for your suggestions. We added the terms.
- Line 29. Please correct “an increased number of dead sperms” by “reduced sperm viability”
Response: We have modified the sentence.
- Line 30. Sperm maturation does not take place in the seminiferous tubules but in the epididymis. Authors must correct this sentence by using “disrupted spermatogenesis” instead of “deterioration of sperm maturation in the seminiferous tubules”.
Response: We have changed the sentence as you pointed out. We apologize for the confusion.
- Line 31. “Fibrosis of the stroma”. Which stroma did the authors refer? Testicular stroma, epididymal stroma or both? Please, specify.
Response: Sorry for the confusion. We had corrected the sentence.
- Line 31. “destructed spermatogenesis was found in the epididymis”. As indicated before, the spermatogenesis takes place in the seminiferous tubules, whereas the sperm maturation occurs in the epididymis. So, authors must correct “denaturated spermatogenesis” by “defective sperm maturation”. By the way, how did the authors analyse the epididymal sperm maturation? The procedure used is not described in the Material and Methods section. Therefore, authors must be careful with the terms used.
Response: Sorry for the confusion. We had modified the sentence, from “denatured spermatogenesis” to “defective sperm maturation”. Unfortunately, we could not analyze the epididymal sperm maturation. We had modified the sentence.
- Lines 32-33. This sentence is unclear, considering that in fact the authors are confuse about where the spermatogenesis and sperm maturation occur.
Response: Thank you for your great suggestion. We had corrected the sentence.
- Lines 36-37. Authors must revise this conclusion because it does not seem obvious from their results.
Response: In the dermatitis model mice, disrupted spermatogenesis was observed, mainly due to the increased inflammatory cytokines derived from the skin lesions. The administration of representative skin-derived cytokines also reproduced the decreased sperm viability. It is critical problem in the clinical field. We have modified the text. Thank you for your suggestion.
Introduction
- Lines 50-52. Which the message? How is this related with the manuscript? Please, revise this sentence in order to provide a clear message according with the topic of the manuscript.
Response: We here showed the research topics for sperm dysfunction as the introduction.
Lines 79. Please, correct “testicular toxicity” by “testicular function”.
Response: We had corrected the text.
Line 80. The authors also analysed the sperm viability. So, it must be added in this sentence.
Response: We had corrected the text.
Results
- As indicated in the general comments, this section must be carefully revised in order to improve its readability; authors must also use the past form in the whole section.
Response: We appreciated your suggestions. We apologized for these errors. We had modified it.
- Lines 153-155. The content of these lines does not belong to the Results section, and they must be eliminated. Please, not that the term “dead sperm” is not correct and authors must refer as “non-viable sperm” when describing the procedure to analyse the sperm viability. Authors should revise the whole document in order to unify the terminology.
Response: Thank you very much for your suggestions. We have deleted the sentences in the results section, and moved to material and method section.
- Line 162. “Sperm viability rate”. Did the authors measure the sperm viability rate? If so, specify how it was calculated. Otherwise, please just refer to “Sperm viability”.
Response: We had modified the term.
- Lines 164-165. It is not necessary to indicate in the Figure legend the statistical test used. So, the content of this line must be removed.
Response: We had removed the sentence.
- Line 172. “no clear change”. What do the authors refer? Please, provide a comprehensive description of the results.
Response: Sorry for the lack of words. No clear change means no abnormal finding of the spermatogenesis in the seminiferous tubules. We have changed the sentence.
- Lines 180-181. “normal maturation process”. How was it evaluated? Please, give some details about the sperm maturation process in wild type mice.
Response: Thank you for your suggestions. Sperm undergo maturation as they transit through the caput, corpus and, cauda epididymis. We evaluated HE section of the epididymis: the caput, corpus, and caudal epididymal regions. Compared to the wild type, sperm was few in KCASP1Tg mice histologically. We had corrected the sentences.
- Line 181-182. What do the authors refer when saying that “the sperm stored in the epididymis at 32-week-old 181 KCASP1Tg mice was few”. Did they perform any comparisons with wild type males?
Response: The sperm stored in the epididymis at 32-week-old KCASP1Tg mice was few compared to wild-type mice at 8, 16, and 32 weeks and also KCASP1Tg mice at 8 and 16 weeks. We had modified the sentences.
- Line 191. Please, eliminate “The” from the beginning of the sentence.
Response: We had removed the word.
- Line 192. Please, correct “Histopathological” by “Histological”. Please, note that the term histopathological is incorrect when referring to wild type samples.
Response: Thank you for your suggestions. We have corrected the sentence.
- Lines 193-200. The information given in these lines is the same as in the results, so it must be improved. Please, note this suggestion was already given in the previous revision.
Response: We have modified the sentences.
- Lines 202 and 205. Please, correct “live cell” by “viable sperm”. Authors must uniform the criteria in the whole document.
Response: Thank you very much for your suggestions. We have modified the sentence.
- Lines 216, 217 and 219. Please, correct “Sperm viability ratio” by “Sperm viability”, since the authors did not calculate this ratio.
Response: We have modified the sentence.
- Line 221. Please, eliminate the last sentence of the figure legend.
Response: We had removed the sentence.
- Lines 224-234. Please, revise the message of these lines in order to provide a clear description of the results obtained. Moreover, avoid starting sentence with “There was”
Response: Thank you for your great suggestion. We have modified the sentences.
- Line 236. Please, eliminate “The” from the title of the figure.
Response: We have removed the word.
- Lines 241. Please, eliminate the last sentence from the Figure legend.
Response: We had removed the sentence.
Discussion
- Some paragraphs of the Discussion are a mere repetition of the Results section; a deep analysis of the significance of testis and epididymis pathologies, as well as of the sperm quality alterations is still lacking.
Response: Thank you for your suggestions. We had modified the discussion section.
- Lines 247-248. Authors must revise the content of these two lines since they did not analyse the semen quality, but the spermatogenesis and sperm maturation. Moreover, the “sperm density” is not a sperm parameter.
Response: Thank you very much for your suggestions. We have changed the sentence.
- Line 276. “Sperm function”. The authors did not analyse the sperm function in their study. I suppose they refer to “sperm motility”. If so, please use appropriate terminology.
Response: Thank you very much for your suggestions. We have modified the term.
Again, “sperm density” is not a sperm parameter. Do the authors refer to sperm concentration? Please, use appropriate terminology. (Line 336)
Response: We apologized. We have corrected the term.
Material and Methods
- Subsection 4.3. Flow cytometry analysis. Authors must indicate: 1) which type of filters did they use to detect the fluorescence emitted by each fluorochrome, 2) how many replicas per sample did they analyse, 3) how many events were analysed by replica.
Response: Thank you for your great suggestion. This information has been supplemented in the 4.3 subsection.
- Subsection 4.3. Flow cytometry analysis. Please, describe the protocol used to analyse the sperm viability.
We appreciated your comments. We added 2.0μl of each dye solution into 2 mL of cell suspension. After vortex and incubation for 5 minutes at room temperature, the cell suspension was analyzed with an Accuri C6. This information has been supplemented in the 4.3 subsection.
- Line 382. Correct “dead sperm” by “non-viable sperm”.
Thank you for your great suggestion.
We have changed from “dead sperm” to “non-viable sperm”.
Reviewer 2 Report
Dear article autors,
I aapreciated your answer to all my questionst. Thank you. I agree with publishing of your article in this repaired form.
Author Response
Comments and Suggestions for Authors
Dear article autors,
I aapreciated your answer to all my questionst. Thank you. I agree with publishing of your article in this repaired form.
Response: We appreciated your comments.
Round 2
Reviewer 1 Report
General comments
Authors have improved the overall quality of the manuscript. Nevertheless, the Discussion section still contains several paragraphs repeating the results obtained and not providing any clue about their significance.
Specific comments
Abstract
- Line 31. “defective sperm maturation was speculated histologically in the epididymis”. Please, provide a more accurate sentence and avoid the use of “speculate”.
Keywords
- Line 37. Please, eliminate “spermatogenesis” from the list, since the study does not include an analysis of the spermatogenesis in any group of animals.
Introduction
- Lines 80-82. Please, revise the content of these lines; there are some grammatical mistakes.
Results
- Line 135. What do the authors refer when saying “the sperm did not function properly”? In fact, authors did not analyse the sperm function, but the sperm motility and sperm viability. The alterations of both parameters may be indicative of alterations in spermatogenesis and/or sperm maturation, or in even of abnormal composition of the seminal plasma. Please, revise this sentence.
- Line 159. What do the authors refer when saying “sperm of KCASP1Tg mice were fragile”. Did the authors measure the sperm fragility? I do not think so. Therefore, I strongly recommend the authors to make their conclusions based on the results obtained from the sperm parameters they analysed.
- Line 170. “no abnormal finding of the spermatogenesis in the seminiferous tubules”. Please, provide evidences that demonstrate that spermatogenesis was not altered. Which were the histological characteristics of the seminiferous epithelium in these males?
- Lines 175-176. “There was no interstitial fibrogenesis in the testes by Masson-trichrome staining of the testis”. Please, revise and correct this sentence.
- Lines 179-181- The terms “abundant” and “few” are too vague. Please provide robust information by analysing the sperm concentration in the epididymis of both group of males.
- Lines 181-182. “The interstitial fibrogenesis was intensively stained in the epididymal stroma 182 by using Masson-trichrome staining”. Please, revise and correct this sentence.
- Line 195. “decreased stored sperm in the seminiferous tubules”. Please refer as “reduced sperm content in the lumen (L) of seminiferous tubules”.
- Lines 195. “L means lumen”. Please, eliminate; the meaning was already indicate in the last comment.
- Lines 195-196. “the sperm in the lumen of KCASP1Tg mouse was large size and denatured”. Please, note that this sentence has no sense; it is necessary to review it in order to provide a clear message.
- Line 198. “There was no abnormal interstitial fibrogenesis in the testes by Masson-trichrome staining of the testis”. Please, revise and correct this sentence.
- Line 200. “the stroma of the periseminiferous tubule”. This is an incorrect term. Authors must refer as “the stroma surrounding the seminiferous tubules”.
- Line 202. “The number of sperm stored in the corpus epididymal regions was few”. After epididymal maturation sperm stores in the epididymal cauda until ejaculation. Authors must revise and correct this sentence.
- Line 209. “a decreased viable sperm”. Please, refer as “decreased sperm viability”.
- Line 212. “decreased the sperm viability”. Please, refer as “decreased sperm viability”.
- Lines 234-237. The content of these lines belongs to the Discussion. Please, remove it from the Results section and add to Discussion.
Discussion
- An analysis of how cytokines affect testicular function is still lacking.
- Authors must review the grammar, in order to ensure that all sentences provide a clear message. Check for instance lines 246-248.
- Line 312. “Sperm is stored in the epididymis”. Please, note that after epididymal maturation sperm is stored in the epididymal cauda until ejaculation.
- Lines 313-314. “The strong fibrogenesis was observed in the epidymis stroma, and environments are not preferable for the stored sperm”. Please, note that the epididymal stroma does not secrete substrates implicated in sperm maturation, but the epithelium. So, authors must carefully review this sentence.
Author Response
General comments
Authors have improved the overall quality of the manuscript. Nevertheless, the Discussion section still contains several paragraphs repeating the results obtained and not providing any clue about their significance.
Response: We appreciated your comments. We have revised the text as you pointed.
However, we received as many as 52 requests from reviewer 1 for the revision, and other 33 requests at the 2nd revision. At the third revision, we receive other 22 requests. Of course, we appreciated several targeted and valuable comments, but we feel something unusual…This cannot be the normal review process. The current reviewer request the expand the discussion section more, but we need to supplement the speculation; however, if we add the speculation the reviewer suggested to avoid the speculation. We need to write the fact.
Specific comments
Abstract
- Line 31. “defective sperm maturation was speculated histologically in the epididymis”. Please, provide a more accurate sentence and avoid the use of “speculate”.
Response: Thank you very much for your suggestions. We have modified the sentence.
Keywords
2. Line 37. Please, eliminate “spermatogenesis” from the list, since the study does not include an analysis of the spermatogenesis in any group of animals.
Response: We appreciated your suggestions. We have deleted the term.
Introduction
3. Lines 80-82. Please, revise the content of these lines; there are some grammatical mistakes.
Response: We appreciated your suggestions. We had modified it.
Results
4. Line 135. What do the authors refer when saying “the sperm did not function properly”? In fact, authors did not analyse the sperm function, but the sperm motility and sperm viability. The alterations of both parameters may be indicative of alterations in spermatogenesis and/or sperm maturation, or in even of abnormal composition of the seminal plasma. Please, revise this sentence.
Response: Thank you very much for your suggestions. We have modified the sentence.
5. Line 159. What do the authors refer when saying “sperm of KCASP1Tg mice were fragile”. Did the authors measure the sperm fragility? I do not think so. Therefore, I strongly recommend the authors to make their conclusions based on the results obtained from the sperm parameters they analysed.
Response: Thank you for your suggestions. We have deleted the term of fragile.
6. Line 170. “no abnormal finding of the spermatogenesis in the seminiferous tubules”. Please, provide evidences that demonstrate that spermatogenesis was not altered. Which were the histological characteristics of the seminiferous epithelium in these males?
Response: We appreciated your suggestions. We have changed the sentence.
Please also refer to the paragraph 2 in the result session 2.5.
7. Lines 175-176. “There was no interstitial fibrogenesis in the testes by Masson-trichrome staining of the testis”. Please, revise and correct this sentence.
Response: We had revised and corrected the sentence.
8. Lines 179-181- The terms “abundant” and “few” are too vague. Please provide robust information by analysing the sperm concentration in the epididymis of both group of males.
Response: Thank you very much for your suggestions. We had changed the terms.
9. Lines 181-182. “The interstitial fibrogenesis was intensively stained in the epididymal stroma 182 by using Masson-trichrome staining”. Please, revise and correct this sentence.
Response: We had revised and corrected this sentence.
10. Line 195. “decreased stored sperm in the seminiferous tubules”. Please refer as “reduced sperm content in the lumen (L) of seminiferous tubules”.
Response: Thank you for your suggestion. We had corrected the sentence.
11. Lines 195. “L means lumen”. Please, eliminate; the meaning was already indicate in the last comment.
Response: Thank you very much for your suggestions. We have deleted the sentence.
12. Lines 195-196. “the sperm in the lumen of KCASP1Tg mouse was large size and denatured”. Please, note that this sentence has no sense; it is necessary to review it in order to provide a clear message.
Response: Thank you very much for your suggestions. We have deleted the sentences, and also changed the images.
13. Line 198. “There was no abnormal interstitial fibrogenesis in the testes by Masson-trichrome staining of the testis”. Please, revise and correct this sentence.
Response: We had corrected this sentence.
14. Line 200. “the stroma of the periseminiferous tubule”. This is an incorrect term. Authors must refer as “the stroma surrounding the seminiferous tubules”.
Response: Thank you very much for your suggestions. We have modified the sentence.
15. Line 202. “The number of sperm stored in the corpus epididymal regions was few”. After epididymal maturation sperm stores in the epididymal cauda until ejaculation. Authors must revise and correct this sentence.
Response: We had revised and corrected this sentence.
16. Line 209. “a decreased viable sperm”. Please, refer as “decreased sperm viability”.
Response: Thank you for your suggestions. We have corrected the sentence.
17. Line 212. “decreased the sperm viability”. Please, refer as “decreased sperm viability”.
Response: Thank you very much for your suggestions. We have modified the sentence.
18. Lines 234-237. The content of these lines belongs to the Discussion. Please, remove it from the Results section and add to Discussion.
Response: Thank you very much for your suggestions. We have deleted the sentences and moved to discussion section.
Discussion
19. An analysis of how cytokines affect testicular function is still lacking.
Response: Thank you very much for your suggestions. We've changed and added some text.
20. Authors must review the grammar, in order to ensure that all sentences provide a clear message. Check for instance lines 246-248.
Response: Thank you very much for your suggestions. We have modified the sentence.
21. Line 312. “Sperm is stored in the epididymis”. Please, note that after epididymal maturation sperm is stored in the epididymal cauda until ejaculation.
Response: We have modified the sentence.
22. Lines 313-314. “The strong fibrogenesis was observed in the epidymis stroma, and environments are not preferable for the stored sperm”. Please, note that the epididymal stroma does not secrete substrates implicated in sperm maturation, but the epithelium. So, authors must carefully review this sentence.
Response: Thank you for your great suggestion. We had corrected the sentence.
Round 3
Reviewer 1 Report
Authors corrected the manuscript according to the comments and suggestions made by this reviewer. Therefore, it can be accepted for publication after minor spelling and grammar corrections.
This manuscript is a resubmission of an earlier submission. The following is a list of the peer review reports and author responses from that submission.
Round 1
Reviewer 1 Report
General comments
This is a study about the relationship between skin inflammation and male infertility using the mouse as an animal model. Despite the topic being of interest, I have some doubts about the experimental design and some of the sperm variables measured. Moreover, the quality of all the sections of the manuscript is low. Therefore, in the Material and Methods section the essential information about the techniques, the number of samples and number of replicas per sample, and the procedure to measure the different variables is lacking. The Results section is not accurate and need to include the mean ± SD of the variables measured and indicate if the differences between groups are either statistically significant or not. The Discussion section is a simple repetition of the results obtained; authors do not provide any clue of how the increased cytokine levels can affect testicular and epididymal function. Moreover, a clear conclusion about the contribution of the study in understanding the relationship between male infertility and skin inflammation is lacking.
Specific comments
Introduction
- Line 49. Authors must change this sentence because the use of the expression “healthy sperm production nor dysfunction of sperm” is not appropriate and it provides a wrong message.
- Lines 59-60. Please, eliminate bold from the text.
Results
- Authors must revise the whole section and use the past form.
- Subsection 2.1. Did the authors perform any statistical analysis to compare the results between both groups? Please, indicate if the differences are either statistically significant or not. Authors must also provide the relative frequencies in order to better compare the results between both groups.
- Subsection 2.2. Did the authors perform any statistical analysis to compare the results between both groups? Please, indicate if the differences are either statistically significant or not.
- Line 85. Authors must indicate if the differences were or were not significant from a statistical point of view and avoid using the terms “apparent” or “noticeable” which are ambiguous.
- Line 86. Please, provide the mean ± SD of the testis size and weight in each group of males at 8, 16 and 32 weeks of age. Indicate if the differences between groups are either statistically significant or not.
- Line 89. Again, authors must compare and clearly indicate if the results obtained are or are not statistically different between both groups. Did the authors calculate the mean ± SD in each group? This is an interesting result that could improve the understanding of the text.
- Lines 93-94. Please, clearly indicate if the differences are statistically significant or not.
- Section 2.3. Authors must provide the mean ± SD of the number of sperm and sperm motility for each male group and week of age.
- Line 98. Which is the relevance of including a video of the sperm movement? Is there any difference in the sperm movement characteristics between both groups of males? Why the video of the sperm movement was not also recorded at 8 and 32 weeks of age?
- Line 101. Authors must indicate if the decrease is statistically significant or not.
- Line 105. “clearly decreased”. What does it mean? Significantly decreased?
- Line 105. “Sperm motility ratio”. Authors must indicate in parenthesis how this ratio was calculated. Please give the results as the mean ± Why did the authors give the sperm motility ration instead of the progressive motility?
- Line 107. The term “sperm were growing” is totally unappropriated. I suppose that the authors refer to that the sperm concentration is low. Please, use appropriate terminology.
- Line 110. “The number of sperm was low and movement was slow in KCASP1Tg mouse”. Did the authors analyse the sperm motility in only one male? Which kind of comparisons did the authors perform to conclude that both the number of sperm and the movement were low?
- Subsection 2.4. Sperm mortality is not a currently used parameter of sperm quality, but sperm viability. So, authors must rewrite this subsection by giving the results of sperm viability. Please provide the mean ± SD of each group of males and indicate if the differences among them are either significant or not. Authors must also consider in expressing the results in percentage instead of total number.
- Lines 129-130. Did the authors measure the generation of reactive oxygen species? If they did, please provide the results obtained. It they did not, please eliminate this sentence. Maybe it could be an important topic for the Discussion.
- Line 133. “in a time-dependent manner”. Please, specify. Do the authors refer from 8 to 32 weeks of age?
- Lines 133. “the number of sperm stored”. Did the authors count the number of spermatozoa stored in the epididymis? If so, please give the results obtained for each group of males and each group of age.
- Subsection 2.5. Please, provide a detailed description of the structural features of the testis and epididymis at 8, 16 and 32 weeks of age in each male group using the information given by H-E and Masson-Trichrome stains. Complete this description with the specific information obtained from HIF1 and IgG stains.
- Line 139. Authors must change the title of the Figure legend. Do the authors consider that the images from the wild type mice are pathological?
- Line 143. Did the authors perform a statistical study to determine if the histological differences between both groups are significant?
- Line 145. Did the authors count the number of sperm cells stored in the epididymis? If so, please give the results obtained in mean ± SD in the text not in the figure legend.
- Figure 6. The legend is a repetition of the content of subsection 2.5. Authors must correct it.
- Line 150. Please, refer to “live” or “viable” cell population. As indicated in a previous comment the sperm mortality is not appropriate sperm parameter, but sperm viability.
- Line 153. “significant difference was undetected”. Does it mean that the differences are not statistically significant? Please, note that this is not appropriate way to express whether the differences between groups are or are not statistically significant.
- Line 155. “Reproduction of sperm fragility”. Please, note that this term does not exist; in fact, the authors give the results of dead sperm population. So, the must change the Figure 7 by giving the results of viable sperm population, and correct both the figure legend and the whole 2.6. subsection.
- Subsection 2.7. The authors must improve the description of the results obtained in order to give a clear message.
- Line 161. The authors must show the cytokine concentration at 8 and 32 weeks of age in both groups. These results will provide an interesting view about the changes in the cytokine content through the time.
- Lines 166-167. Please revise the sentences and clearly indicate if the differences between groups were either significant or non-significant.
- Lines 169-172. The information given in the legend of Figure 8 is the same as that given in the text. Please, correct.
Discussion
- Lines 176-177. “number of produced sperm”. What do the authors refer? The sperm concentration or the rate of spermatogenesis? Did the authors measure any of these parameters? Please, clarify. If so, please, include the results obtained in the Results section
- Line 177. “the sperm did not function properly”. The authors did not measure the sperm function, but the sperm viability and sperm motility. Therefore, they can only conclude that it leads to a reduced sperm viability and motility.
- Lines 196-205. The content of these lines is a mere repetition of the results, and they do not provide any interpretation about the causes of testicular size reduction and decreased sperm counts and sperm viability. Please, improve it.
- Line 204. The authors did not measure the “sperm fragility”, so avoid the use of this term.
- Lines 206-216. Authors must improve the content of this paragraph by clearly correlating how cytokines may induce Sertoli cell detachment from the basal lamina and the disturbances in spermatogenesis.
- Lines 214-216. In these lines the authors just repeat the results obtained without giving any relationship between cytokines and epididymal maturation. Please, also note that if the rate of spermatogenesis is reduced the number of spermatozoa stored in the epididymis also decreases. So, authors must be careful when discussing the effects of cytokines in testis and epididymis function.
- Line 219. As previously indicated, the appropriate sperm parameter is sperm viability.
- Line 220. Again, the authors did not measure the “sperm fragility”.
- Lines 219-231. The content of these lines is a mere repetition of the results obtained. Authors do not provide any clue about the effects of cytokines on testicular function and on sperm viability and sperm motility.
- Lines 232-235. The content of these lines cannot be considered as a conclusion of the work. Authors must clearly indicate the relevance of their results and their contribution in the understanding of how skin inflammation can affect male infertility.
Material and Methods
- Lines 246-247. Please, indicate clearly the number of males used.
- Line 250-251. Again, indicate the number of males used.
- Subsection 4.2. Lines 266-267. I have some concerns about the sperm motility ratio. The sperm motility is usually expressed as the percentage of total motile spermatozoa and the percentage of progressive motile spermatozoa of the whole sperm population; so, I encourage the authors to express the values in %. Please, also indicate the number of animals used and the number of replicas analysed per male,
- Subsection 4.3. Line 269. The sperm count is usually given as number of spermatozoa/ml. Please, correct it.
- Lines 272. Please, provide a better description of the affinity of thiazole orange, as well as some references of other studies using this procedure to measure the sperm viability.
- Subsection 4.5. Please, provide more information about the procedure used, as well as some references of other studies using it.
- Subsection 4.6. The description of the Statistical analysis is too much vague. Please, provide a more detailed information. Which were the variables? And the treatments? Did the authors test for normality and homoscedasticity of the variables?
References
- The manuscript contents only 15 references, and just a couple of them were related with the effects of cytokines in testis and epididymis funcion. Authors must both expand the reference list and reinforce the Discussion by comparing their results with those obtained by other autors in similar studies.
- The reference style is not uniformed.
Reviewer 2 Report
How you can explain different sperm count using Makler sperm count chamber and Flow cytometry analysis (FCM) in 32 weeks old mice in Fig.4A and B?
The biggest difference was seen in 16 weeks old mice in both approaches in sperm count but then sperm motility rate different the most in 32 weeks old mice. I think it will be great to discuss more deeply why we can see this effect and wha it can happen.
Technical note:
The citation no. 11 in not stated in bold. Is it correct?
There are some missing gaps after words in sentences in part 2.7.
Overall it seems to be great work/paper done on the field of research in reproduction and causes of intertility.